# The Role of Amino Acid Metabolism of Tumor Associated Macrophages in the Development of Colorectal Cancer

**DOI:** 10.3390/cells11244106

**Published:** 2022-12-17

**Authors:** Manman Jiang, Hongquan Cui, Zhihong Liu, Xin Zhou, Ling Zhang, Longnv Cao, Miao Wang

**Affiliations:** 1Suzhou Center for Disease Control, Suzhou 214000, China; 2Oncology Department Suzhou Kowloon Hospital, Shanghai Jiaotong University School of Medcine, Suzhou 214000, China; 3The Second Hospital of Soochow University, Soochow University, Suzhou 214000, China; 4Oncology Department, Siyang Hospital, Suqian 223798, China

**Keywords:** colorectal cancer, tumor microenvironment (TME), amino acid metabolism, tumor associated macrophages (TAM), cell death

## Abstract

Tumor-associated macrophages (TAMs) are important immune cells in the tumor microenvironment (TME). Previous studies have shown that TAMs play a dual role in the development of colorectal cancer and promote the additional exploration of the immune escape of colorectal cancer. Studies have confirmed that macrophages utilize amino acid metabolism under the stimulation of some factors released by tumor cells, thus affecting the direction of polarization. Therefore, we investigated the effect of amino acid metabolism on macrophage function and the involved mechanism. Based on the comprehensive analysis of the GSE18804 GEO dataset and amino acid metabolism pathway, we identified the eight key enzymes of amino acid metabolism in colon TAMs, namely, ACADM, ACADS, GPX4, GSR, HADH, HMGCL, HMGCS1 and IDH1. We then evaluated the expression, survival analysis and relationship of clinicopathological features with these eight key enzymes. The results supported the critical role of these eight genes in colorectal cancer. Macrophages phagocytose tumor cells, and these eight key enzymes were identified in combination with GPX4, a critical protein of ferroptosis, suggesting that the change in the expression of these eight key enzymes in TAMs may be involved in the regulation of colorectal cancer through cell death. Correlation analysis of three programmed cell death (PCD) marker genes indicated that these eight key enzymes may cause macrophage death through pyroptosis, leading to immune escape of colorectal cancer. We also investigated the regulation of ACADS in CRC using flow cytometry, qPCR and ELISAs, which demonstrated that an ACADS deficiency polarizes TAMs to M2 macrophages. In summary, the present study revealed the relationship between amino acid metabolism and the cell death of macrophages, providing a new research direction for the molecular mechanism of macrophage polarization.

## 1. Introduction

Colorectal cancer (CRC) is one of the most prevalent malignant tumors of the lower gastrointestinal tract and is closely related to factors such as genetics, diet and work habits [1]. According to the latest statistics, the number of patients who die of colorectal cancer accounts for 9% of the total cancer deaths every year [2]. Although patients who die from colorectal cancer are mainly older, patients with CRC have gradually tended to become younger with recent changes in diet and irregular lifestyles [3]. At present, the first-line treatment of CRC is surgical resection, supplemented by the combination of radiotherapy and chemotherapy [4]. However, the five-year survival rate of postoperative patients is poor, indicating the need for the development of effective methods to prevent and treat colorectal cancer early.

The tumor microenvironment (TME) has been a hot area of research in recent years with the aim of identifying new therapeutic breakthroughs to meet the challenges of cancer treatment [5]. In addition to the major tumor cells, the TME is comprised of cancer-associated fibroblasts (CAFS), abundant blood vessels and various types of immune cells, all of which play an important role in the reproductive and metastatic processes of tumors [6]. The particular metabolic activities of tumor cells cause a hypoxic and weakly acidic state in the tumor entity, which not only facilitates their own growth and proliferation, but also inhibits the proliferation and function of antitumor immune cells [7]. Previously, most studies have been focused on direct tumor-killing immune cells, such as CD8 and NK cells, but the associated clinical prognosis remains unsatisfactory [8,9]. In recent years, further understanding of the tumor microenvironment has led research to discover that tumors are infiltrated with a large number of procancer immune cells, such as TAMs, mostly manifesting as M2-type macrophages, in addition to beneficial antitumor immune cells [10]. Therefore, the reverse thinking of inhibiting cancer-promoting immune cells to achieve better and faster treatment, has become a new breakthrough in cancer research.

Macrophages are a type of phagocyte, a critical component of the intrinsic immune system, and they have a strong ability to phagocytose and kill pathogens, thereby participating in the inflammatory response and immunomodulatory effects [11]. Macrophages originate from myeloid stem cells, differentiate from monocytes, are located in systemic tissues and are endowed with specific physiological functions according to needs [12]. Macrophages are polarized by different stimulation conditions into predominantly M1-type and M2-type macrophages [13]. Interestingly, both types of macrophages are polarized from M0-type macrophages but have different physiological functions and metabolic patterns [13]. M1 macrophages obtain energy mainly through glycolysis, and secrete large amounts of proinflammatory factors and tumor necrosis factors; in addition, they have biological functions to promote inflammation and inhibit cancer [13]. M2 macrophages use oxidative phosphorylation to supply energy [14]. Due to their release of anti-inflammatory factors, chemokines and growth factors, most of M2 macrophages have anti-inflammatory effects and promote the occurrence and development of tumors [15]. The hypoxic environment in the tumor microenvironment provides favorable conditions for the polarization of TAMs to M2 macrophages, which in turn produce vascular growth factors and chemokines that promote cancer cell reproduction and metastasis, while regulating and inhibiting the antitumor effects of other immune cells.

The initial goal of the present study was to change this unfavorable trend and provide new research options for colorectal cancer treatment and prognosis by identifying amino acid-metabolizing enzymes that regulate the polarization of tumor-associated macrophages to M2 macrophages. Using database shared resources, we compared and identified many genes with significant differences in expression between TAMs infiltrated by colorectal cancer and monocytes in blood; this suggested that TAM metabolism was strongly linked to its own amino acid metabolic enzymes. We screened 16 key enzymes involved in amino acid metabolism from these changed genes. We then evaluated the relationship of these 16 enzymes with tumor invasion and the prognosis of colorectal cancer, and we determined that there were eight amino acid enzymes that have a key effect on the prognosis and staging of colorectal cancer.: ACADM, ACADS, GPX4, GSR, HADH, HMGCL, HMGCS1 and IDH1. We also analyzed a public database and found that the levels of these eight genes may affect the invasion and metastasis of colorectal tumors, and we selected the rare and representative ACADS gene for experimental validation. The results demonstrated that the tumor of ACADS-deficient mice grew faster and the number of M2 macrophages infiltrating the tumor increased. Combined with bioinformatics analysis, we hypothesized that the loss of ACADS regulates the metabolism of tumor-associated macrophages and promotes M2 macrophage polarization, resulting in antitumor immunity to promote the occurrence and development of CRC.

## 2. Materials and Methods

### 2.1. Identification of Genes Associated with TAM in CRC

The microarray data of TAMs in CRC were downloaded from the GSE18804 GEO database (http://www.ncbi.nlm.nih.gov/geo, accessed on 3 August 2022), which is one of the largest repositories containing thousands of array- and sequence-based data in the world. The GSE18804 dataset includes global gene expression data of colorectal cancer-associated macrophages (colon TAMs), brain tumor-associated macrophages (brain TAMs) and control macrophages. Colon TAMs and brain TAMs were induced in vitro by incubation with soluble factors released from HT29 or U-87MG cells. The genes that were expressed at different levels in colon TAMs and control macrophages may be associated with TAMs in CRC.

### 2.2. Identification of Enzymes Involved in Amino Acid Metabolism

The diagram of the amino acid metabolic pathway was downloaded from WikiPathways [16] (https://www.wikipathways.org/, 3 August 2022) [17], which is an open platform dedicated to the administration of biological pathways for everyone. We selected 126 amino acid metabolic enzymes from the diagram of the amino acid metabolic pathway.

### 2.3. Identification of Key Enzymes for Amino Acid Metabolism in Colon TAMs

Using our identified TAM-related genes and amino acid metabolism-related enzymes, we predicted that the genes present in both datasets are likely to be key enzymes for amino acid metabolism in colon TAMs. Based on the results of the survival analysis of these key genes in colorectal cancer, we screened for TAM amino acid-metabolizing enzymes that are critical for the development of colorectal cancer. All data for survival analysis were obtained from the GEPIA2 database (http://gepia2.cancer-pku.cn/, 3 August 2022) [18], which is a one-stop platform for multiple analysis and visualization, containing the RNA sequencing expression data of 9736 tumors and 8587 normal samples from TCGA and GTEx projects. The statistical methods to assess the prognostic impact of genes in all survival analyses were performed in COAD using the “Overall Survival” function.

### 2.4. Expression of Critical Genes in COAD

UALCAN (http://ualcan.path.uab.edu/, 3 August 2022) [19] is a powerful online tool for visualizing and presenting TCGA data. In the present study, we retrieved and visualized the expression of critical genes in colorectal cancer and adjacent normal colorectal tissues by UALCAN.

### 2.5. Clinicopathological Analysis of Critical Genes in COAD

As mentioned previously, GEPIA2 is a comprehensive platform containing a variety of user-friendly analyses. We used the “Stage Plot” of “Expression DIY” to evaluate the relationship between the mRNA expression of critical genes in COAD and individual cancer stages.

### 2.6. Correlation Analysis of Critical Genes

LinkedOmics (http://www.linkedomics.org/, 3 August 2022) [20] includes multiomics data from all 32 TCGA Cancer types and 10 Clinical Proteomics Tumor Analysis Consortium (CPTAC) cancer cohorts. In the present study, we used the “LinkFinder” analytical module to obtain associations between the screened critical genes and biomarkers of necroptosis, pyroptosis and ferroptosis. For the correlation analysis, we used the Pearson test. Then we showed the results via heat maps (“*****” represents *p* value, color represents Signal Strength).

### 2.7. In Vivo Tumorigenesis

All the animal procedures were approved by the Institutional Animal Care and Use Committee of Soochow University (Suzhou, China; approval number SYXK2017-0043).

Six to eight-week-old Balb/c and Balb/cByJ male mice were purchased from the Jackson Laboratory (Bar Harbor, ME, USA). We injected 2 × 10^6^ CT26 cells/0.1 mL subcutaneously into the back of each Balb/c and Balb/cByJ mouse (6 mice in each group and 12 mice in total). After 5 days, we measured the volume of tumors on the backs of mice with a Vernier caliper every other day, and we counted and recorded them. On Day 17, because the volume of the largest tumor was close to the maximum standard (diameter < 20 mm), we sacrificed all mice with cervical dislocation and removed all tumors for measurement and recording. All removed tumors were photographed and archived, followed by subsequent experimental studies.

### 2.8. Flow Cytometry

First, we collected the tumor tissues of the two groups and lysed them with a prewarmed dissociation buffer (1 mg/mL collagenase I and 20 µg/mL DNase I). After 30 min, the lysate was filtered through a 70-µm cell strainer after the termination of digestion with a DMEM high glucose medium (containing 5% FBS). Cells were then blocked with Fc Block CD16/CD32 (1:50 dilution for a concentration of 0.5 µg per well) and stained with the antibodies in a staining buffer (1% BSA): anti-F4/80, anti-CD86 and anti-CD206 (BD Biosciences, CA, USA, 1:5000). Finally, the Canto II flow cytometry system and FlowJo_V10 software were used for flow cytometry analysis. 

### 2.9. Real-Time PCR

Total RNA from mouse subcutaneous tumor tissue was isolated by TRIzol reagent (Invitrogen, Carlsbad, CA, USA) and reverse transcribed into cDNA using RT Master Mix for qPCR (HY-K0510, MedChemExpress, NJ, USA). SYBR Premix Ex Taq™ (TaKaRa, Kyoto, Japan) was used for a quantitative real-time polymerase chain reaction (qRT-PCR) with GAPDH as the reference gene for expression analysis. The primer sequences are shown in Table 1.

### 2.10. Enzyme-Linked Immunosorbent Assay (ELISA)

The concentrations of TNF-α, IL-10, TGF-β and Gzmb in the supernatant of subcutaneous tumor grafts from the two groups of mice were measured separately using commercial ELISA kits (Sigma-Aldrich, St. Louis, MO, USA) according to the manufacturer’s instructions (see Appendix A).

### 2.11. Statistical Analysis

Data are presented as the means ± standard deviations. The Student’s *t* test was performed using GraphPad Prism version 8.0 software (GraphPad Software Inc., San Diego, CA, USA). All figures were created in R software (version 4.0.4). The criterion for statistical significance was *p* < 0.05.

## 3. Results

### 3.1. Identification of Key Enzymes for Amino Acid Metabolism in Colon TAMs

We used the mRNA expression levels of colon TAMs and control macrophages from the GSE18804 dataset to screen for genes associated with TAMs in colorectal cancer. Based on the criteria of *p* < 0.01, there were 918 genes with elevated expression and 829 genes with reduced expression in colon TAMs, compared to the control macrophages (Figure 1A). Therefore, these 1747 genes were considered to be genes associated with TAMs in CRC. Using the cellular metabolic pathway diagram from WikiPathways, we obtained the key enzymes involved in cellular amino acid metabolism (Figure 1B). As these genes were present in both groups, they were considered as TAM metabolism-related genes. There were 16 overlapping genes in the two groups, as presented in the Venn diagram (Figure 1C). Although all 16 genes were closely related to amino acid metabolism in TAMs, only the expression level of a gene that can affect the survival of colorectal cancer patients should be considered a key enzyme for amino acid metabolism in colon TAMs. Consequently, we retrieved the survival analysis data of these 16 genes in GEPIA2, and the elevated expression level of 8 genes led to a better prognosis for colorectal cancer patients: ACADM, ACADS, GPX4, GSR, HADH, HMGCL, HMGCS1 and IDH1 (*p* < 0.05) (Figure 1D). 

### 3.2. The Expression of Most of the Key Enzymes Was Downregulated in CRC

Because we screened these eight genes as key enzymes for amino acid metabolism in colon TAMs, their expression in colorectal cancer tissues should be significantly different from that in the paired normal colorectal tissues. We used the UALCAN database to show the expression of these eight genes in colorectal cancer and normal colorectal tissues (Figure 2A–H). As expected, the outcomes showed that the expression of five genes (ACADM, ACADS, GSR, HADH and HMGCL) in CRC was lower than that in paired normal colorectal tissues (*p* < 0.001). In addition, the average expression level of IDH1 in colorectal cancer tissues was lower than its average expression level in normal colorectal tissues, but this difference was not statistically significant due to the large individual differences (Figure 2H). Unexpectedly, the expression of GPX4 (Figure 2C) and HMGCS1 (Figure 2G) in colorectal cancer was upregulated.

### 3.3. Low Expression of Key Enzymes Is Associated with Adverse Clinicopathological Features of CRC

The patient’s stage often indicates the progression and prognosis of cancer, and the classification criteria for cancer staging are generally based on the clinicopathological features of the patient. Therefore, we explored the relationship between the expression levels of these key enzymes and the clinicopathological features of patients with colorectal cancer by GEPIA2. Except for GPX4 (Figure 3C) and HMGCS1 (Figure 3G), the results were consistent with the expression levels in cancer tissues and normal colorectal tissues, as the expression levels decreased with the deterioration of clinicopathological conditions in patients with colorectal cancer (Figure 3). Interestingly, the decreased expression levels of ACADM, GSR and HADH were closely related to the deterioration of colorectal cancer patients (*p* < 0.01). Therefore, these findings suggested that these three genes may be the key enzymes regulating the development of colorectal cancer.

### 3.4. The Expression of Key Enzymes Is Closely Related to Cell Death

In recent years, necroptosis, pyroptosis and ferroptosis have increasingly become the focus of attention in the field of tumor research, and we previously identified five to six markers of each death mode [21]. By searching the correlation between key enzymes and markers in LinkedOmics, we evaluated the relationship between key enzymes and cell death. The correlation heatmap showed that these eight key enzymes were closely related to pyroptosis, suggesting that the changes in amino acid metabolism of colon TAMs may affect the development of tumors through cell pyroptosis (Figure 4B). Furthermore, the correlation of ACADS, GSR and HADH with cellular pyroptosis was prominent, and the correlation of ACADS, GSR and HMGCL showed a strong correlation with cellular necroptosis (Figure 4A). For ferroptosis, ACADM, GPX4 and GSR were most likely to be involved in the regulatory process, with GPX4 being an important biomarker for the occurrence of ferroptosis (Figure 4C).

### 3.5. The Deletion of ACADS Promotes Tumor Growth and Induces TAM Polarization to M2 Macrophages

Because Balb/cByJ mice can spontaneously produce a 276-bp deletion in the structural gene of ACADS [22], we regarded Balb/cByJ mice as ACADS -/- mice. By subcutaneously injecting colorectal cancer cells (CT26) into Balb/c and Balb/cByJ mice to construct a mouse subcutaneous tumor model, we observed that the deletion of ACADS in mice resulted in the significantly accelerated growth of colorectal cancer (Figure 5A). We removed the tumors and dissociated the cells for flow cytometry analysis, qRT-PCR analysis and ELISAs. The proportion of M2 macrophages in tumor tissue was significantly higher in Balb/cByJ mice than in Balb/c mice, as shown by flow cytometry analysis (Figure 5B,C); this suggested that the deletion of ACADS could induce TAM polarization to M2 macrophages. Finally, the qRT-PCR (Figure 5D) and ELISA (Figure 5E) results confirmed these findings, as the expression of M2 macrophage markers in tumor tissues was significantly higher in Balb/cByJ mice than in Balb/c mice, both at the mRNA and protein levels.

## 4. Discussion

The tumor microenvironment (TME), including the complex extracellular matrix and various stromal cells, is the environment in which tumor cells survive. Among the components of the TME, tumor-associated macrophages (TAMs), as the most abundant immune cell type in the TME, regulate tumorigenesis and development through multiple pathways [23]. There are two types of macrophages, namely, M1 macrophages and M2 macrophages. Although M1 macrophages and M2 macrophages are of the same origin, they play opposite roles in tumor development, promoting the study of TAM polarization as the focus of tumor immunotherapy [24]. Increasing evidence suggests that M1 macrophages and M2 macrophages are distinctly different in terms of the mode and level of cellular metabolism, particularly amino acid metabolism [25,26]. Amino acid metabolism is an important part of supporting life activities. Moreover, immune cells are dependent on amino acid metabolism for energy and biomass [27]. Glutamine metabolism [28] and arginine metabolism [29] have been reported to support and regulate macrophages. The mechanisms of amino acid metabolism, affecting the function of TAMs in regulating tumor development, has been a focus of research for tumor immunotherapy targeting TAMs. The aim of the present study was to identify the key enzymes of amino acid metabolism in TAMs and explore their functions and mechanisms in tumors. In addition to our previous experience with colorectal cancer research [30,31], we selected colorectal cancer as the direction of the present study because it has a high incidence of malignant tumors, causing 940,000 deaths worldwide every year [2]. A critical precancerous lesion of colorectal cancer is inflammatory bowel disease, which is largely driven by macrophages, in terms of pathophysiological mechanisms. 

Based on transcriptome sequencing data (GSE18804) from The University of Tokyo, we analyzed and obtained differential genes for colon TAMs. Based on these differentially expressed genes, we selected 16 enzymes involved in amino acid metabolism and as key genes involved in the differentiation of monocytes to TAMs. Although all 16 enzymes may play a role in TAM differentiation, they may not necessarily have a significant impact on the development of colorectal cancer. Therefore, we used the correlation of the expression level of genes with the prognosis of colorectal cancer patients as an evaluation index, and finally identified eight critical enzymes of amino acid metabolism in colon TAMs, namely, ACADM, ACADS, GPX4, GSR, HADH, HMGCL, HMGCS1 and IDH1. To further verify the key role of these eight genes, we examined their expression in colorectal cancer and their impact on tumor invasion and metastasis. As expected, the results indicated that most of these eight amino acid-metabolizing enzymes in TAMs, except for GPX4 and HMGSC1, have a significant effect on the development and progression of colorectal cancer. HMGCS1 is a critical ketogenic enzyme in the human body. Lin et al. [32] confirmed that knockdown of HMGCS1 does not decrease the protein expression level of HMGCS1, but instead causes a compensatory increase in the expression of HMGCL, indicating that the expression of HMGCS1 and its correlation with clinicopathological features should be opposite to that of HMGCL; this would be consistent with the results of our analysis. It is well known that the inhibition of GPX4 function is a critical aspect in the occurrence of ferroptosis. In addition, phagocytosis is the basic function of macrophages [13]. Consequently, we speculated that these eight key enzymes may play a role in regulating colorectal cancer through cell death.

Programmed cell death (PCD) comes in many forms and is a self-protective mechanism of the organism. Recent studies have revealed that necroptosis, pyroptosis and ferroptosis have a tremendous impact on tumor and immune cells [33,34,35]. Necroptosis is a form of cell death in which PRK1 and RIPK3 assemble into an oligomeric complex (necrosome), which, in turn, causes cell lysis via the mixed-lineage kinase domain-like protein (MLKL) [36]. Pyroptosis is a GSDMD- or GSDME-dependent cell death activated by caspases 1/3/4/5, and cells that die in this manner release a variety of proinflammatory factors, including IL-1β, IL-18 and HMGB1 [37]. Ferroptosis is a newly identified iron-dependent PCD, induced by lipid peroxidation accumulation that eventually activates ACSL4 activation and inactivates GPX4 [38]. To explore whether the selected eight key enzymes play a role through these three PCDs, we evaluated the relationship between these eight genes and various biomarkers [21] of necroptosis, pyroptosis and ferroptosis in colorectal cancer. Interestingly, these eight genes were closely related to three kinds of PCD, and the correlation with pyroptosis was the most prominent. Therefore, we hypothesized that the altered amino acid metabolism of TAMs makes them more prone to pyroptosis, preventing their involvement in phagocytosis of tumor cells and allowing tumors to grow rapidly. As this is the most innovative discovery of our research, it will be our future research direction.

Based on all the analyses in the present study, ACADS was the most notable of these eight key enzymes, in terms of its impact on prognosis and clinicopathological staging, as well as its association with cell death pathways. Therefore, we further explored the regulatory mechanism of ACADS on TAMs in CRC. The subcutaneous injection of colorectal cancer cells into Balb/c and Balb/cByJ mice demonstrated that the deletion of ACADS promoted the growth of colorectal cancer. We also analyzed the tumors of both groups by flow cytometry, qPCR and ELISA, which demonstrated that ACADS deficiency promoted the polarization of TAMs to M2 macrophages at the cellular level or at the mRNA and protein levels. The present findings elucidated a new mechanism of ACADS in regulating the immune escape of colorectal cancer.

## 5. Conclusions

In summary, the present study identified eight key enzymes of amino acid metabolism in TAMs that are involved in the immune escape process of colorectal cancer through PCD, providing a new molecular mechanism for TAM polarization in colorectal cancer. Nevertheless, the present study had certain limitations. The specific regulatory mechanism of these eight key enzymes in TAMs and their specific application in the clinical treatment of colorectal cancer need to be further explored.

## Figures and Tables

**Figure 1 cells-11-04106-f001:**
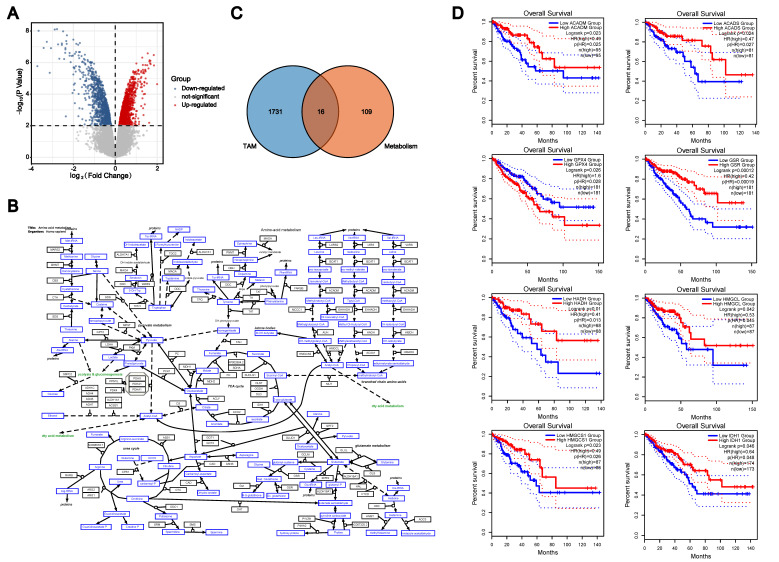
Identification of key enzymes for amino acid metabolism in colon TAMs. (**A**). The mRNA expression levels of colon TAMs and control macrophages from the GSE18804 dataset. (**B**). The cellular metabolic pathway diagram from WikiPathways. (**C**). The Venn diagram of TAM−related genes and amino acid metabolism-related enzymes. (**D**). The survival analysis data of these eight genes in GEPIA2.

**Figure 2 cells-11-04106-f002:**
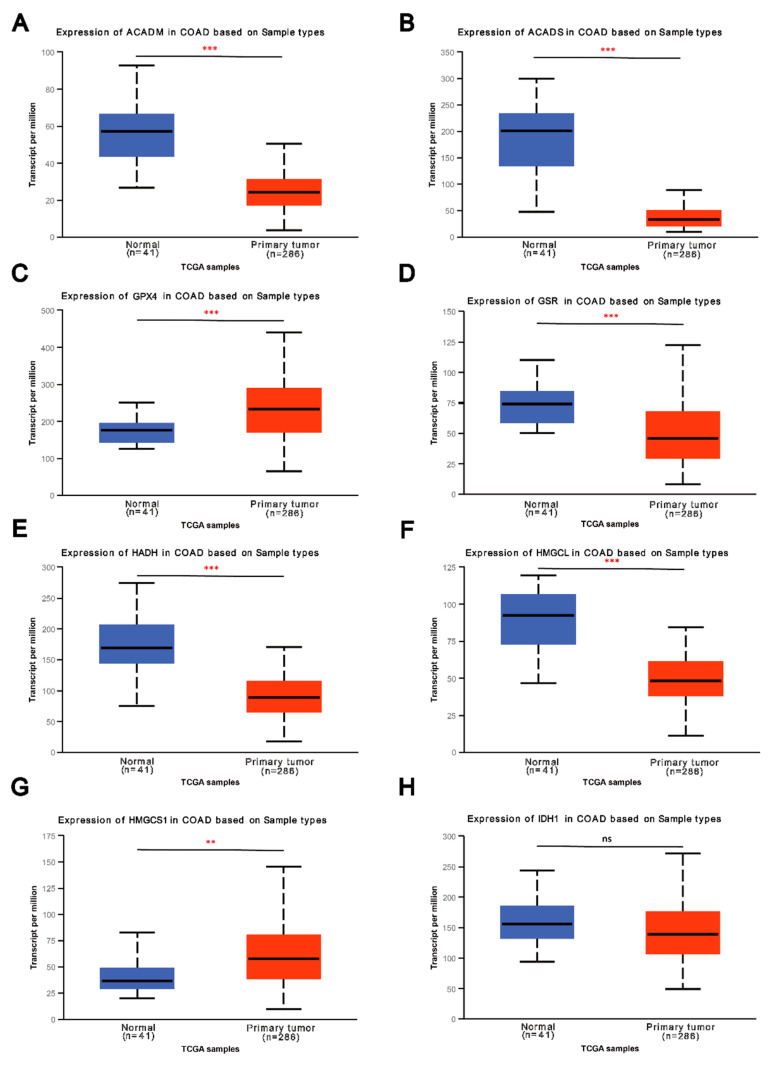
The expression of most of the key enzymes was downregulated in CRC. (**A**–**H**). The expression of these eight genes in colorectal cancer and normal colorectal tissues from UALCAN. (“**ns**” represents no significance, “******” represents *p < 0.01*, “*******” represents *p<0.001*). The expression of five genes (ACADM, ACADS, GSR, HADH and HMGCL) in CRC were extremely lower than those in paired normal colorectal tissues (*p* < 0.001).

**Figure 3 cells-11-04106-f003:**
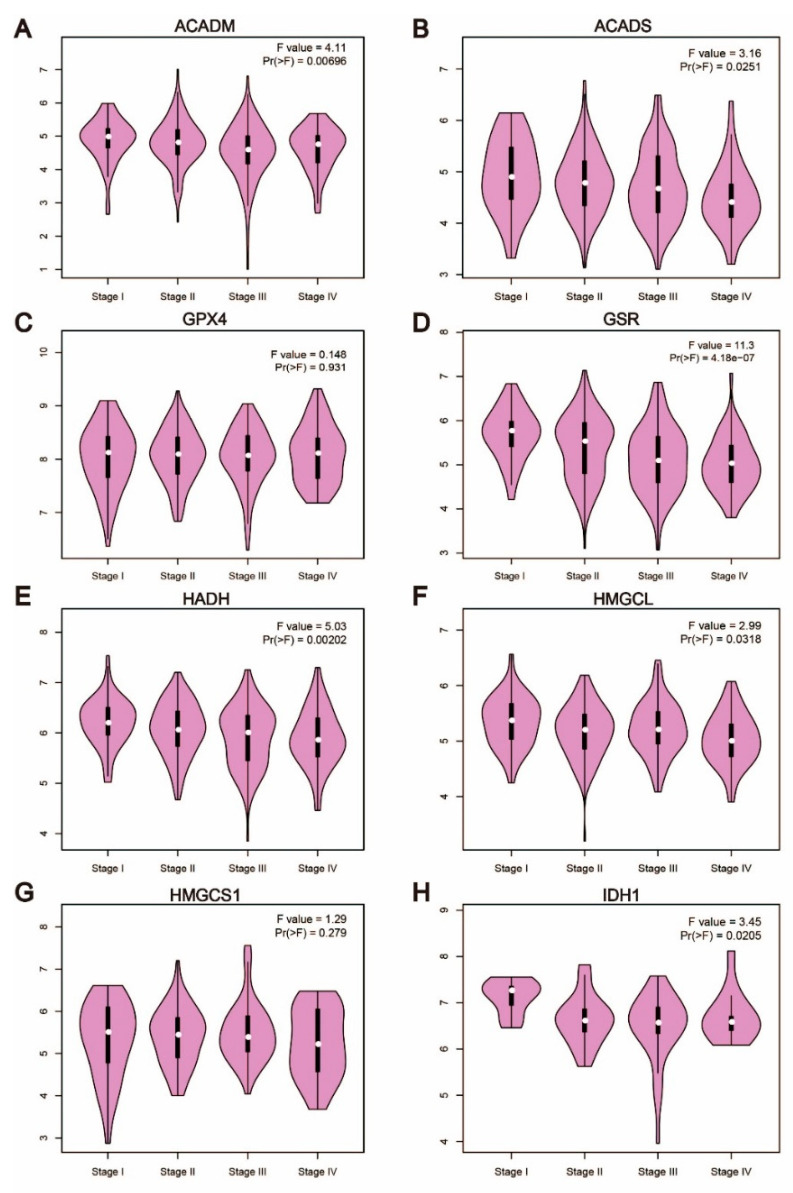
Low expression of key enzymes is associated with adverse clinicopathological features of CRC. (**A**–**H**). The relationship between the expression of these enzymes and the clinicopathological features of patients with colorectal cancer by GEPIA2. Apart from GPX4 and HMGCS1, the expression levels of the other six genes decreased with the deterioration of clinicopathological conditions in patients with colorectal cancer.

**Figure 4 cells-11-04106-f004:**
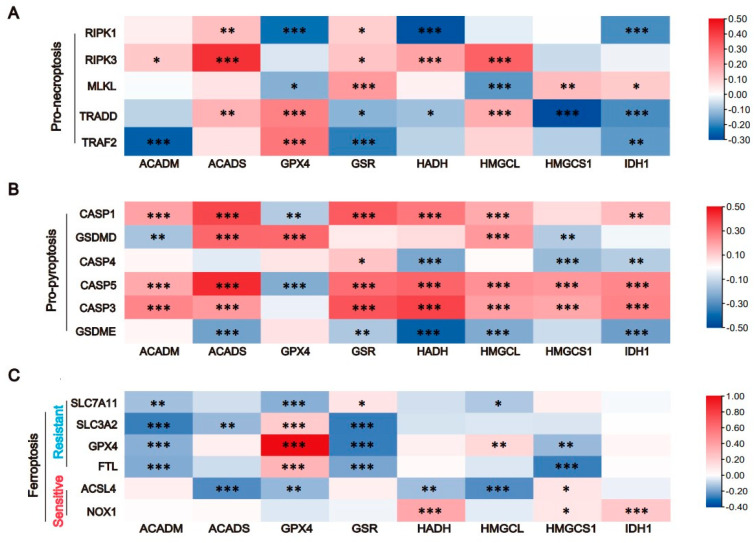
The expression of key enzymes is closely related to cell death. (**A**). The correlation of these eight genes with cellular necroptosis. (**B**). The correlation of these eight genes with cellular pyroptosis. (**C**). The correlation of these eight genes with cellular ferroptosis. (“*****” represents *p* < 0.05, “******” represents *p* < 0.01, “*******” represents *p* < 0.001).

**Figure 5 cells-11-04106-f005:**
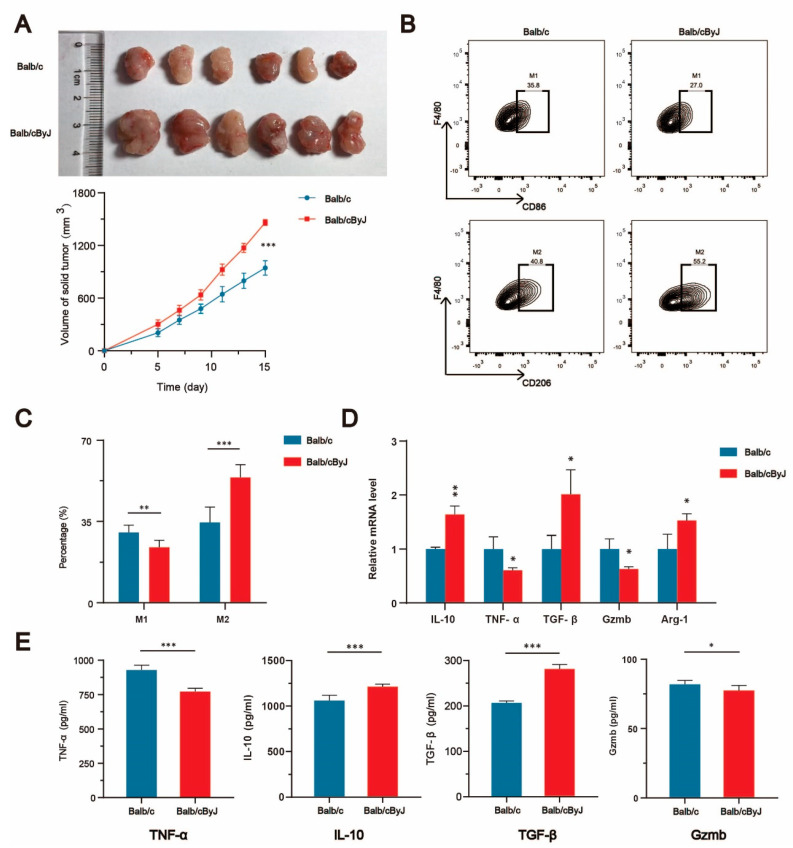
The deletion of ACADS promotes tumor growth and induces TAM polarization to M2 macrophages**.** (**A**). Mouse subcutaneous tumor model in Balb/c and Balb/cByJ mice. (**B**,**C**). Flow cytometry analysis of TAMs. (**D**). The expression of factors related with TAMs in tumor tissues was detected by qPCR. (**E**). The expression of factors related with TAMs in tumor tissues was detected by ELISA. (“*****” represents *p* < 0.05, “******” represents *p* < 0.01, “*******” represents *p* < 0.001).

**Table 1 cells-11-04106-t001:** Forward and reverse primers used in qPCR experiments.

Genes	Forward Primer	Reverse Primer
IL-10	5′-GCCATGAATGAATTTGACA-3′	5′-CAAGGAGTTGTTTCCGTTA-3′
TNF-α	5′-ACCACCATCAAGGACTCAA-3′	5′-CAGGGAAGAATCTGGAAAG-3′
TGF-β	5′-ATTCCTGGCGTTACCTTGG-3′	5′-AGCCCTGTATTCCGTCTCCT-3′
Gzmb	5′-CCTGCTACTGCTGACCTTGT-3′	5′-AGGCTGCTGATCCTTGATCG-3′
Arg-1	5′-CAGAAGAATGGAAGAGTCAG-3′	5′-CAGATATGCAGGGAGTC-3′
GAPDH	5′-GGTCCCAGCTTAGGTTCAT-3	5′-CAATCTCCACTTTGCCACT-3

## Data Availability

The data used in the current study are available from the corresponding author on reasonable request.

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
