# Peer review of "The Role of Amino Acid Metabolism of Tumor Associated Macrophages in the Development of Colorectal Cancer"

_cells, 2022, doi:10.3390/cells11244106_

Round 1
Reviewer 1 Report
In this paper by Manman Jiang et al, the authors developed a retrospective study looking for different metabolic enzymes associated with macrophage metabolism during the different staging of colorectal cancer. Next, the authors tried to associate their in-silico findings with a murine in vivo model, where they found some markers associated with M2 macrophage development. Overall, it is an interesting paper with new findings. However, the author must achieve several major points before accepting this manuscript.
Major points:
Regarding the in-silico analysis, it seems with no problems at all, However, some figures are impossible to see. For example, Fig1B is not possible to read. So, the reader cannot determine whether the finding is good.
Figure 2 Please indicate any significant value, if there is no significant differences is not worth doing a graph.
In figure 3, please keep the same scale on the x-axis to better appreciate any difference among the groups.
Figure 4 needs a better explanation; what do the different amounts of * mean?
Figure 5, is Gnzb granzyme B, if it is, it seems very close between BALB/c and BALB/cByJ, and is Granzyme B an M2 marker? Please explain.
How long were the subcutaneous grafts cultured? Which culture media was used?
Were the cytokines detected were by spontaneous release or the cells or tissues were stimulated? Is granzyme B a factor released by macrophages?
Why do the authors not evaluate metabolic enzymes in the in vivo model? It is mandatory if the authors want to validate their in-silico findings.
A clear marker for M2 macrophages is the expression of PDL1 and /or PDL2, as well as mannose receptors. Thus, these markers must be evaluated to support their claims.
There are many acronyms that need to be spelled out at least once.
The data on macrophages from the in vivo model is unclear, given that the authors are using a whole tumor and the cytokines may come from different types of cells recruited to the tumor microenvironment. To support their claims, the authors must isolate macrophages (cell-sorting) and perform the experiment on such isolated cells.
Reviewer 2 Report
In the current study, the authors primarily used public databases to examine the association between changes in gene expression in tumor-associated macrophages and survival in colorectal cancer. First, the authors showed that the expression of five genes, ACADM, ACADS, GSR, HADH and HMGCL, is downregulated in tumor macrophages in colorectal cancer. The authors showed that reduced expression levels of ACADM, GSR, and HADH are closely associated with the deterioration of patients with colorectal cancer. Thus, these findings suggest that these three genes may be important enzymes that regulate the development of colorectal cancer. Finally, the authors found that loss of ACADS promotes tumor growth and induces polarity of TAMs to M2 macrophages in a mouse model. Although the manuscript is potentially interesting, the authors have not presented results that fully support their conclusions and the manuscript is too preliminarily in its current stage. Specific comments are as follows.
Major points.
1. The authors should examine whether these genes are actually altered in TAMs in mouse models of colorectal cancer and clarify whether they are consistent with public databases.
2. The authors have shown that d colon cancer growth was inhibited in ACADS-deficient mice (Balb/cByJ mice), but this has already been reported by other investigators. Therefore, Most of the data in this paper is not novel and the authors should provide new data themselves.
Minor points.
1. English should be carefully revised by a professional English editing service.
2. Figure 1 (especially for Figure 1B) is too small and I cannot see well.
3. For Figure 4, the authors provide little explanation of how the data were collected. The methods should be described in detail so that other researchers can replicate them.
Reviewer 3 Report
This manuscript concludes that eight key enzymes in TAMs may be involved in the regulation of colorectal cancer through macrophage death via pyroptosis, leading to immune escape
M1 macrophages obtain energy mainly through glycolysis and secrete large amounts of proinflammatory factors and tumor necrosis factors as well as biological functions to promote inflammation and inhibit cancer. M2 macrophages use oxidative phosphorylation to supply energy.
Due to their release of anti-inflammatory factors, chemokines, and growth factors, M2 macrophages have anti-inflammatory effects and promote the occurrence and development of tumors.
Combined with bioinformatics analysis, they hypothesized that the loss of ACADS regulates the metabolism of tumor-associated macrophages and promotes M2 macrophage polarization, resulting in antitumor immunity to promote the occurrence and development of CRC.
Colon TAMs and brain TAMs were induced in vitro by incubation with soluble factors released from HT29 or U-87MG cells.
As expected, the outcomes showed that the expression of five genes (ACADM, ACADS, GSR, HADH and HMGCL) in CRC was lower than that in paired normal tissues (P<0.001). Unexpectedly, the expression of GPX4 (Figure 2C) and HMGCS1 (Figure 2G) in colorectal cancer was upregulated. This finding was unexplained and didn’t correspond with the main hypothesis. It needs further explanation.
The mRNA expression data were derived from large databank analysis. While this is a powerful screening tool, the threshold of p<0.01 may not be stringent enough. Furthermore, using the published survival data the metabolic enzymes identified as significant were only tested at the p<0.05 level. Once again, this is a very generous threshold given the variability in the source data. Please justify this.
Although all 16 enzymes that were identified as differentially expressed may play a role in TAM differentiation, they may not necessarily have a significant impact on the development of colorectal cancer. Therefore, they used the correlation of the expression level of genes with the prognosis of colorectal cancer patients as an evaluation index and finally identified eight critical enzymes of amino acid metabolism in colon TAMs. This effort to justify the significance of the findings is a bit far-fetched without a causative relationship. It needs better justification.
The relationship between these metabolically important enzymes and the etiology of CRC is speculative and not proven in this investigation. The authors try to associate the enzyme levels with clinical stage and prognosis, M2 macrophage differentiation and levels, and potential for immune escape of the tumors. While these associations may be interesting, they are not mechanistic or causative, and they are of course not on the same patient population. This relationship is necessary but not sufficient to prove causation.
The authors stress the relationship between inflammation and the development of CRC. “a critical precancerous lesion of colorectal cancer is inflammatory bowel disease, which is largely driven by macrophages in generating polyps.” While this is true, that inflammatory precursor lesions produce a small fraction of CR cancers, and they probably develop and progress through a different mechanism characterized by Vogelstein’s paradigm. Thus the inflammatory precursor may be different than that of conventional CRC. This needs to be stressed in all aspects of the manuscript.
The in vivo work is quite limited, with only 12 animals. Furthermore, the subcutaneous injection of the tumors is, at best, a preliminary representation of the tumors in a xenograft model. This unfortunately raises questions about how the results are representative of the natural situation and the generalizability of the results.
The description of the control macrophages is not clear. Were they the induced colon or brain TAM’s? This needs a better description and potentially better selection of a comparison group. Why should they be the comparison group?
The adjacent “normal tissue” warrants a better characterization.
Minor editorial corrections such as p3, section 2.7 “2x106” are needed.
Reviewer 4 Report
The manuscript is poorly drafted. There are grammatical errors throughout the manuscript.
Modify the title as the in vivo work focuses on ACADS which is a crucial for fatty acid metabolism. Although it plays role in amino acid metabolism too, but the title needs to be more precise.
Provide a explanation for choosing ACADS gene for this study over 8 other enzymes.
Line- 41. Provide CRC in the beginning of paragraph, Line 37.
Line 43. Use CRC instead of colorectal carcinoma.
Line 62: Rephrase this line to avoid repetition of words.
Line 67: mention the different stimulus for polarization of macrophages and add reference here.
Line 70: Add reference for this statement ‘M1 macrophages obtain energy……’
Line 73: Due to release of anti-inflammatory factors…authors are suggested to modify this line as there is one subset of M2 macrophages i.e M2b which are known to produce pro inflammatory cytokines like IL-1, IL6 etc.
Line 105-108: Describe the generation of colon TAMs and brain TAMs in detail mentioning the time for induction, soluble factors specifically used for induction.
Method 2.1. Line- 105-108. Did you perform this experiment? Please clarify the methods in detail. It is confusing.
Line 112: What is the basis for selection of 126 amino acids metabolic enzymes from that pathway.
Line 143: Why authors have chosen only male mice for this study?
Line 144: Add reference for line we injected …….Balb/cByJ mouse.
Line 158: Authors are suggested to add reference for flow cytometry methodology, also mention the concentration/dilution of antibodies used for this study.
Line 159: correct the line “Finally we a canto…..”
Line 167: Add reference for the primers used or if designed by then give details.
Line 169: Authors are suggested to provide a detail methodology for ELISA like the source of sample, procuring, processing of sample for elisa.
Figure 1B- Quality is not good. Provide figure with good resolution.
Figure 5B has only FACS plots. Please provide analysis through bar graphs and significance.
Line 183: Authors have compared colon TAM to the control macrophages. What are those control macrophages?
Line 260: Authors have characterized the macrophage population from tumor. They are advised to use one or more markers to characterize M1 and M2 population.
Authors have shown the role of ACADS in TAM polarization. Did they check any effect of ACADS deletion on normal macrophage polarization status?
What is the mRNA expression level of CD206. Authors are advised to check the M1 markers at mRNA level and protein levels too.
Round 2
Reviewer 2 Report
The authors have addressed almost all of my concerns and I have no further comments.